# DEEP ECOLOGICAL INFERENCE

## ABSTRACT

We introduce an efficient approximation to the loss function for the ecological inference problem, where individual labels are predicted from aggregates. This allows us to construct ecological versions of linear models, deep neural networks, and Bayesian neural networks. Using these models we infer probabilities of vote choice for candidates in the Maryland 2018 midterm elections for 2,322,277 voters in 2055 precincts. We show that increased network depth and joint learning of multiple races within an election improves the accuracy of ecological inference when compared to benchmark data from polling. Additionally we leverage data on the joint distribution of ballots (available from ballot images which are public for election administration purposes) to show that joint learning leads to significantly improved recovery of the covariance structure for multi-task ecological inference. Our approach also allows learning latent representations of voters, which we show outperform raw demographics for leave-one-out prediction.

## 1  INTRODUCTION

Ecological inference (EI), or learning labels from label proportions, is the problem of trying to make predictions about individual units from observations about aggregates. The canonical case is voting. We cannot observe individual people's votes, but people live in precincts, and we know for each precinct what the final vote count was. The problem is to try to estimate probabilities that a particular type of individual voted for a candidate.

Since we can not observe individual labels, but only sums of pre-specified groups of labels, non-identifiability is inherent to the ecological inference problem. The possibility of interaction effects between any relevant demographics and the aggregation groups themselves also means that Simpson's paradox type confounding is an ever present risk. The most basic approach to this problem involves assuming total heterogeneity at the precinct level, and simply assigning the final distribution of votes in a precinct to each person living in that precinct. However, typically people are sorted geographically along characteristics that are politically salient, and that variation can be leveraged to learn information about voting patterns based on those demographics. Classical ecological regressions use aggregate demographics, but here we have access to individual-level demographics via a commercial voter file with individual records, and therefore we construct our models at the individual level. There are a number of advantages to using individual demographics for ecological inference, but note that while individual-level features are observed, individual-level labels still can not be observed, and therefore the fundamental challenges of non-identifiability and aggregation paradoxes remain.

**Related Work**  Classical ecological inference typically assumes an underlying individual linear model and constructs estimators for those model coefficients using aggregated demographics and labels King (1997). More recent work has used distribution regression for large-scale ecological inference incorporating Census microdata in nationwide elections in the US Flaxman et al. (2016). Aggregated labels represent a substantial loss of information that could be used to constrain inferences, and all ecological methods rely on the analyst making assumptions which are not definitively empirically testable from those aggregates alone. Some research has been done on visual techniques for determining when some of these assumptions may have been violated Gelman et al. (2001). Other work has sought to impose additional constraints on the ecological problem by incorporating information from multiple elections Park et al. (2014), which also has the benefit of allowing for estimation of voter transitions, which themselves are of interest.

Our models are built from individual records of commercial voter file data, which leads to some immediate differences between classical methods. First, we do not need to determine the composition of the electorate from an ecological model, since we already know who from the total population voted from administrative records and we only need to use an ecological model for vote choice (who votes is public). Second, we can directly specify a model for vote choice at the individual level and train this using a suitable ecological loss function, as opposed to specifying a model only on aggregates. This is convenient and allows the analyst a great deal of flexibility in modeling choices, and opens the possibility of using ecological inference for individual-level vote choice prediction as well as individual-level latent space modeling. However it does not automatically resolve the fundamental difficulties of ecological methods. We therefore introduce additional constraints on the ecological problem by jointly training on multiple races in a given election year in a manner analogous to existing methods, but adapted to our individual-level framework.

**Contributions**  We develop a loss function that allows us to approximate the poisson binomial and poisson multinomial loss that sits at the center of the individually-oriented approach to EI, but are too intractable to optimize directly. This approach allows us to extend EI with deep learning, providing three key benefits: learning non-linear relationships in data, jointly learning multiple aggregated outcomes and learning low-dimensional high-information representations of individual voters. We apply these methods to data from the Maryland 2018 midterm election, to estimate vote propensity in the elections for the Governor, US Senator, and Attorney General. We validate these estimates using three datasets: first, post-election survey data on individuals' vote choices; second, data on the joint distribution of candidate support for these three races; and third we validate the learned representations by predicting individual responses to surveys using data linked to the representation.

## 2 An Approximate Ecological Loss and Architectures

### 2.1 Approximating the Poisson Binomial and Multinomial Loss

We model the choice of candidate (including abstaining from voting on a particular race) as a realization from a categorical distribution, according to an individually defined vector of probabilities $p^{(i)}$, so that each individuals "vote" is a one-hot encoded vector, where the indicator represents the candidate the selected. If individuals are modeled as independent realizations of non-identically distributed categorical variables then precincts are distributed as poisson multinomial variables, parameterized by a matrix $P$ where each row corresponds to an individual's set of probabilities. The likelihood of the observed precinct-level counts is then:

$$\mathcal{L}(v_1, v_2, \ldots, v_C | \mathbf{P}) = \log \left( \sum_{A \in F_C} \prod_{(i,c(i)) \in A} P_{c(i)}^{(i)} \right) \tag{1}$$

Where $F_C$ is the set of all possible assignments $(i, c(i))$ of an individual $i$ to a vote choice $c(i)$, subject to the constraint that the total number of individuals assigned to each candidate $c$ is $v_c$, so that $|F_C| = \frac{(\sum_{c=1}^{C} v_c)!}{\prod_{c=1}^{C} v_c!}$. Even for the simplest possible case of a single race with only two candidates, this loss is not straightforward to compute Hong (2013). In practice this loss is intractable to compute for the voting setting, where $C > 3$ (the United States has a two party vote system, and people need not fill vote in every race at the ballot box) and $N > 20,000$. We propose to approximate this loss by assuming as

$$\mathcal{L}(v_1, v_2, \ldots, v_C | \mathbf{P}) = \log \left( \sum_{A \in F_C} \prod_{(i,c(i)) \in A} \left( \frac{1}{N} \sum_{n=1}^{N} P_{c(i)}^{(n)} \right) \right) \tag{2}$$

$$= \log \left( \left( \prod_{c=1}^{C} \left( \frac{1}{N} \sum_{n=1}^{N} P_c^{(n)} \right)^{v_c} \right) \sum_{A \in F_C} 1 \right) \tag{3}$$

$$= \log \left( \prod_{c=1}^{C} \left( \frac{1}{N} \sum_{n=1}^{N} P_c^{(n)} \right)^{v_c} \right) + \log \left( \frac{(\sum_{c=1}^{C} v_c)!}{\prod_{c=1}^{C} v_c!} \right) \tag{4}$$

Where (4) is the multinomial loss using the precinct-average of the predicted probabilities for each candidate. This approximation is correct for the first moment, but overestimates the variance compared to the variance of the poisson multinomial distribution. It is however extremely efficient to compute for a large number of precincts in parallel. For the poisson binomial we employ a similar strategy.

## 2.2 EXTENDING ECOLOGICAL INFERENCE

### 2.2.1 ARCHITECTURES

We consider two architectures, each of which were implemented using Bayesian and non-Bayesian neural networks. The first architecture is a dense network, and the second is an extension of a varying-intercepts varying-slopes model that has been used for modeling voting behavior in Ghitza & Gelman (2013). A varying-intercepts varying-slopes model is typically written as follows for a binary model (extensions to multiple classes are straightforward)

$$logit(p_i) = \alpha_0 + \sum_{k=1}^{K} \alpha_{k[i]} + \sum_{l=1}^{L} X_{il} \sum_{k=1}^{K} \beta_{k[i]l} \tag{5}$$

Where we have K groups of random effects and L fixed effects. Including a column of ones in X to account for the intercept terms and absorbing the $\beta$s into the $\alpha$s, we can rewrite this as a linear model where the coefficients themselves are computed from another linear model as follows

$$logit(p_i) = \alpha_0 + \sum_{l=1}^{L} X_{il} \gamma_{il} \tag{6}$$

$$\gamma_{il} = \sum_{k=1}^{K} \alpha_{k[i]l} \tag{7}$$

Here, the slopes for the fixed effects are the sum of additive contributions which depend on some categorical variables for unit i. Therefore, $\beta_{il}$ can be interpreted as an embedding of the categorical features, which are learned in such a way that the when taken in a linear combination with the numerical features X, the results give good predictions on the probability scale. Categorical features tend to be sparse, and so the embedding reduces the dimensionality of those input features. This concept can be extended by predicting $\beta_{il}$ from a general embedding of a matrix of categorical features $X^c$ so that

$$\beta_{il} = f(X_i^c) \tag{8}$$

Where in this study, f is given by a fully connected neural network. The advantage of this architecture is that globally it has very high capacity to fit complex data, but locally it has linear structure. We call this architecture a Deep multi-level model (MLM). We find that this leads to more reasonable estimates of partisan crossover voting, that is republican voters voting for Democratic candidates and vice versa, than the dense network, which tends to underestimate the degree of crossover relative to what is found in survey data. This architecture also provides advantages in interpretation because it is fundamentally a linear model.

### 2.2.2 BAYESIAN LAYERS

For the Bayesian implementation for both architectures, we use the following multilevel formulation for each dense layer.

$$\sigma \sim p_\sigma = HalfNormal(0, S) \qquad (9)$$
$$w \sim p_w = Normal(0, \sigma) \qquad (10)$$
$$b \sim p_b = Normal(0, \sigma) \qquad (11)$$
$$y = activation(wx + b) \qquad (12)$$

for layer inputs x and outputs y, where S is a hyperparameter that controls the prior on the regularization strength. These multilevel layers help desensitize the model from our choice of hyperparameters, while allowing for different regularization strengths in each layer. We use a fully factorized mean-field variational approximation with the standard non-centered parameterization.

Bayesian neural nets often under perform as a result of training difficulties that arise from increased variance in the stochastic gradients. To mitigate this, we train all Bayesian nets using KL annealing, where the variational standard deviation parameters are set to small values and the KL divergence term in the variational loss is initially zeroed out, and gradually increased over the course of training. This allows the model to find a solution basin about as quickly as a non Bayesian net, essentially by training the variational mean parameters with the standard deviations fixed. Once inside a basin we optimize the full objective, and this tends to give good results.

## 3 APPLICATION TO THE MARYLAND 2018 MIDTERM ELECTIONS

We applied this method to the statewide races in the Maryland 2018 midterm elections. Maryland typically votes for Democrats but had a Republican governor in 2018 who had high approval ratings going into the race and won reelection by a little less than 300,000 votes. Maryland also had an attorney general and US senate race in that same election, where incumbent Democrats won by large margins. This gives us a useful test case for examining not correlations in voting patterns between races.

We demonstrate two practical use cases for out methods. The first is the classical use case for ecological regression, where the analyst wants to learn demographic breakdowns of the vote share in the election. This involves fitting an ecological model and then cross tabulating the model scores by various quantities of interest. While it is impossible to determine the ground truth for this, we compare the cross tabulations from our ecological model to estimates of support from a large sample survey which is weighted to be representative of Maryland voters. We additionally leverage a dataset we are not aware of having been used in this validation setting: the distribution of ballots is public information. This means if we predict multiple races we can additionally check how well our recovers the joint distribution of votes for candidates in different offices.

The second, which to our knowledge is new for ecological modeling, is to use an ecological model to train a latent space for a few shot learning model, which we then trained on survey data. We show that the features learned by the EI model are more predictive on survey data than the raw features, with significantly reduced dimensionality.

### 3.1 DATA

The data for these models come from two primary sources. The first is Maryland board of elections, which publishes vote tallies at the voting precinct level. The second is a commercial voterfile, which contains administrative data such as addresses (which are used to determine voting precinct assignments), age, party registration, and whether or not each person cast a ballot in the 2018 election. This file is also augmented with model scores provided by political data vendors, which attempt to estimate demographic characteristics such as race and education, as well as political characteristics such as strength of partisanship. We use both the administrative and modeled data as the features for the model, along with additional features consisting of aggregate characteristics of census tracts such as median income and percentage college educated. We use three other datasets for validating the models. Data on how each individual actually votes is obviously not available. We use the

Cooperative Congressional Election Study to get estimates of candidate support in the 2018 Maryland election. This data has been shown to very accurately reflect the election results, making it well suited for validation (Agadjanian & Robinson (2019)). We also want to be able to examine the joint distribution over several races. Information on the joint distribution of ballots is actually public information, and political scientists have recently begun cleaning these data for use in analysis. We actually chose Maryland for this study because such data was available (Kuriwaki (2019); Agadjanian & Robinson (2019)). Finally, we want to be able to estimate the quality of learned latent representations. We use data from polling performed over the course of 6 months by the think tank Data for Progress, where we have 132 Maryland respondents linked to an individual in the voterfile mentioned above. We have a set of 24 questions across all these respondents pertaining to political questions, primarily about partisan affiliation, presidential approval, candidate preference in the 2020 election, and what news sources respondents rely on.

## 3.2 RESULTS

We trained a number of model variants for this study. We trained standard and Bayes nets, dense and MLM architectures with depths ranging from 0 to 5 (specifically 0, 1, 3, 5). We trained all these models jointly and independently for all the races we consider, and with the binomial and multinomial losses. We trained each model variant 5 times and selected the model with the best hold out performance as the representative for that variant. Results presented here are for jointly-trained Bayes nets with binomial loss unless otherwise indicated. We also select the optimal depth on the test $R^2$ unless explicitly plotting over model depth.

First we compare the cross tabulated EI model scores to a post-election survey from the Cooperative Congressional Election Study. This survey asked about the three races we examine here: Attorney General, Governor and US Senate. Generally we see performance improvements from the deep models, and more benefit from the MLM. We see benefits to Bayesian layers here as well, which is clear from Table 1. Individuals have a strong tendency to vote for their co-partisans, and so partisanship is highly predictive of voting. However survey data shows that partisan crossover voting is quite common, and EI appears to have a tendency to underestimate the extent that partisans crossover. Crossover is essentially a matter of capturing the interaction of smaller effects which affect crossover with that of a vary large effect, namely partisanship. We conjecture that the inductive biases of our architecture facilitate the estimation of such interactions.

A major benefit of the neural network framework is the ability to jointly train all the races in an election which share a common latent space. This allows for more accurate estimation of the covariance structure between votes in different races, which is of interest since like partisan crossover: these correlations can explain a great deal about an election. The benefits of joint training are very clear from Figure 3.2. Panel A shows example modeled covariances and the true covariance estimated from the joint ballot results. We see significant benefits to depth for learning the intra-race correlations, but only in the joint learning case. We also find that especially for the MLM joint training improves the demographic vote propensity. Interestingly joint training does not seem to improve demographics estimation for the dense net, which is a bit counter-intuitive because it does improve covariance estimation. This means that at the level of granularity we can validate at the covariance and demographic support metrics are somewhat orthogonal.

Training deep models also has the advantage of allowing us to use the learned latent space in future few shot learning applications. We apply these learned features in a linear model on polling data from Maryland, and show the loo score for different architectures and depths. While our architecture had advantages for predicting partisan crossover, a dense architecture is clearly preferred for the transfer learning application. This is likely because our architecture does not reduce the numerical features. Increased depth and use of Bayes nets has clear advantages for learning predictive features. We find that the Bayes nets tend to be sparser, which leads to better out of distribution predictions for our survey data.

## CONCLUSION AND FUTURE WORK

We presented a framework for ecological inference for cases where individual records are available but only aggregated labels are observed, which is the case for building models of election results from a commercial voter file and voter precinct data. We propose a binomial/multinomial

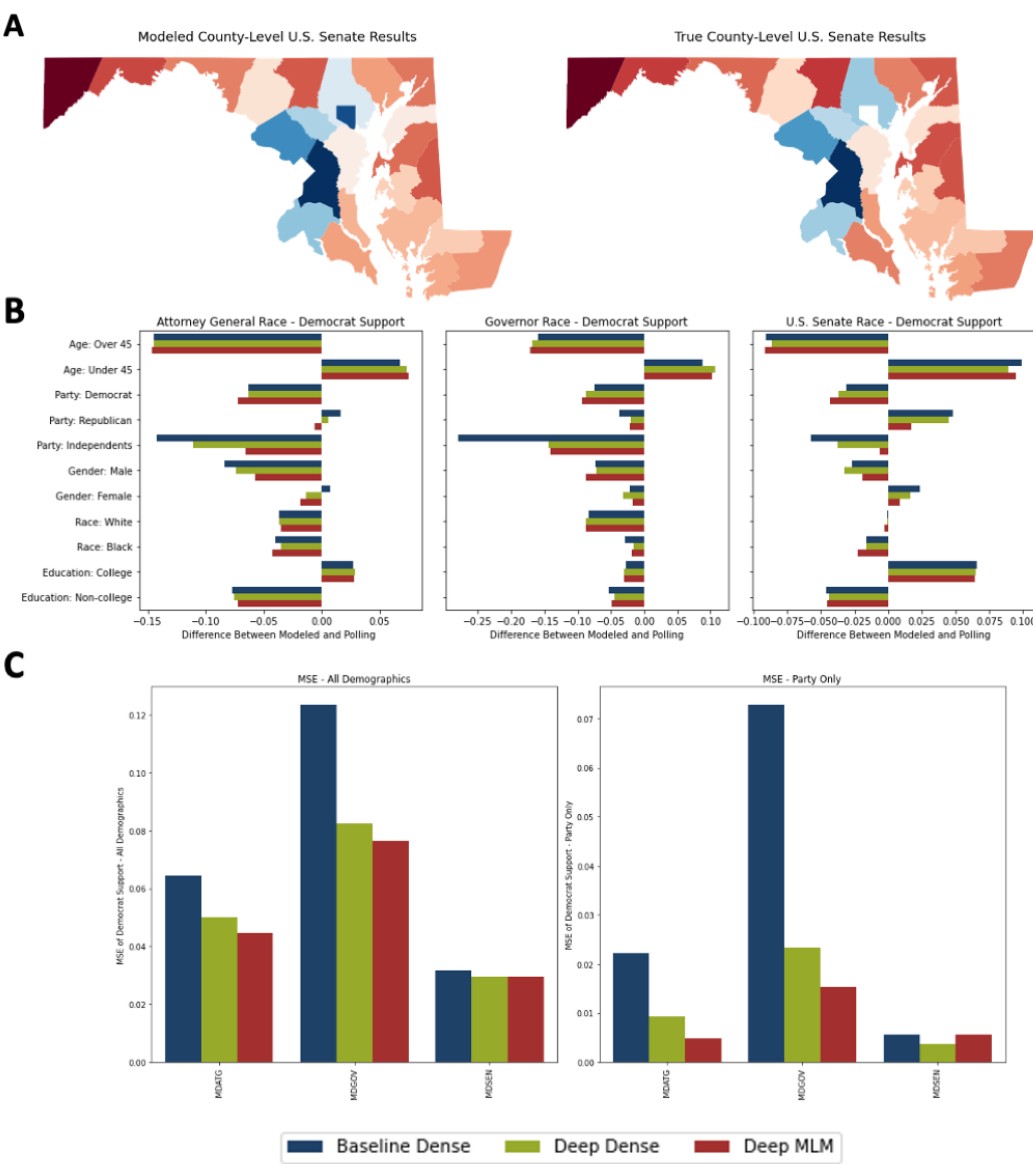

Figure 1: Panel A shows the modeled county-level results from the Deep MLM versus the true county-level results. Panel B shows the difference between modeled and true demographic support levels for three models: the baseline linear (dense) model, the deep dense model, and the deep MLM. Support is a 0-1 scale so that a 0.15 corresponds to modeling 15 percentage points more Democratic support within a demographic than is actually displayed. Panel C is an aggregated representation of Panel B. We show the MSE of support for all demographics and for the three party demographics (where much of the improvement is coming from.

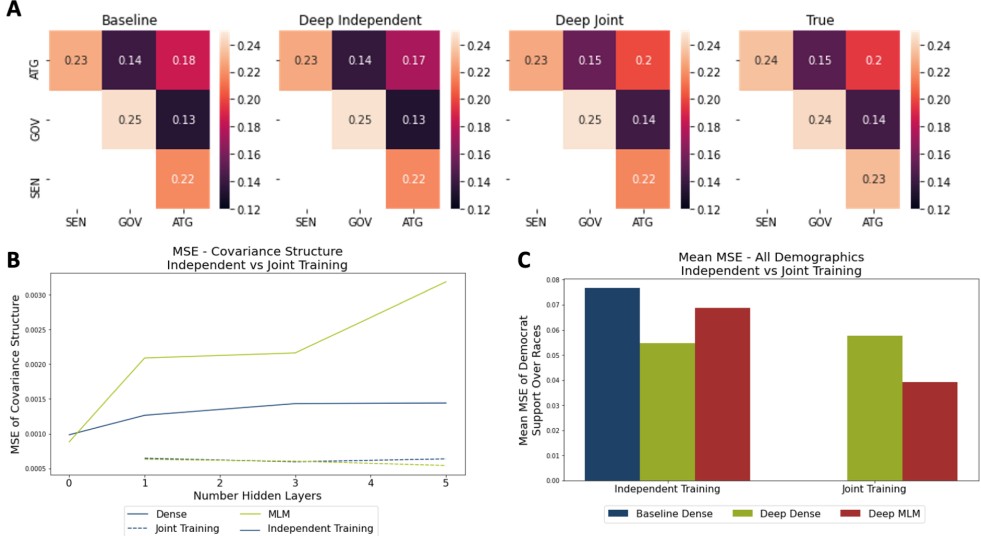

Figure 2: Here we demonstrate the benefits of joint training to learning the covariance structure over races. Panel A shows three modeled covariances: the baseline linear model using our loss function, a deep model trained independently, and a deep model trained jointly. The deep models are the best performing dense nets. Panel B shows that the examples in Panel A are representative. We plot the MSE of the modeled covariance against depth, where the solid lines indicate independent training and dotted lines joint training. We see that joint training leads to significant gains in estimating the covariance. Panel C shows the mean MSE of modeled demographic support over all races. It shows that – especially for the MLM – joint training can improve demograhic-level support estimates.

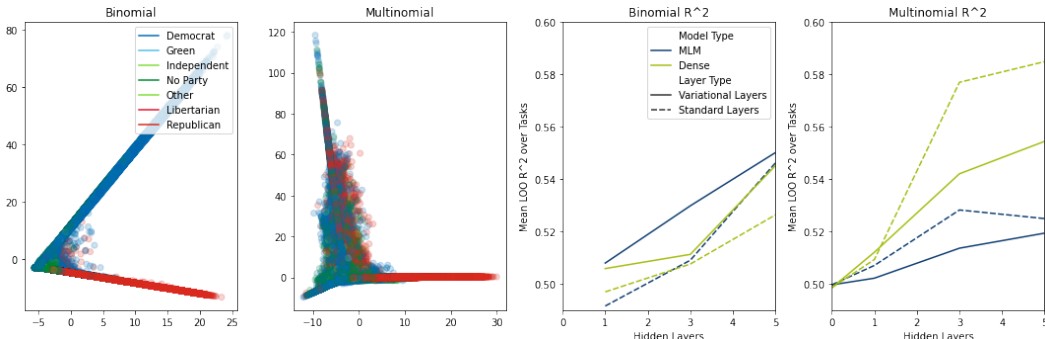

Figure 3: The first two panels here show two dimensional PCA embeddings of the latent representations of all Maryland voters from the best performing binomial and multinomial models, colored by the registered party. The binomial loss embeds voters along a partisanship axis. The multinomial embedding is more three dimensional, reducing less to pure partisanship. The second two panels show the results of training linear models to predict survey responses using the latent space, for latent spaces of various depths. performance of the latent space for predicting survey responses. The metric here is the leave-one-out $R^2$ over all 24 prediction tasks. The x-axis shows the number of hidden layers. The 0 hidden layer "latent space" is just the model input; the raw demographic variables. We see as depth increases so does the quality of the latent space for out of distribution prediction.

| Model | Layer Type | Demo. MSE | Test $R^2$ | Cov. MSE | LOO $R^2$ |
|---|---|---|---|---|---|
| Baseline Dense | Standard | 0.077576 | **0.988824** | 0.000858 | 0.498582 |
| Baseline Dense | Bayes | 0.073171 | 0.988441 | 0.001111 | - |
| Deep Dense | Standard | 0.056831 | 0.988227 | **0.000432** | 0.546267 |
| Deep Dense | Bayes | 0.054882 | 0.986031 | 0.000526 | **0.570651** |
| Deep MLM | Standard | 0.055656 | 0.984454 | 0.000447 | 0.517463 |
| Deep MLM | Bayes | **0.050220** | 0.985576 | 0.000579 | 0.526413 |

Table 1: Results summary table for joint models, comparing Dense and MLM architectures, Standard and Dense layers. Demo. MSE is the mean squared error of the modeled support for the Democrat within a demographic (see Fig. 3.2) over races. Test $R^2$ is the $R^2$ of the predicted precinct level two-way vote share on a held out test set. Cov. MSE is the men squared error of the modeled covariance structure. LOO $R^2$ is the leave-one-out $R^2$ of linear models built using the latent space from the model. For baseline models the "latent space" is simply the raw demographic variables.

loss function with averaged rate parameters as an efficient approximation to the poisson binomial/multinomial. This framework allows for straightforward application of a large family of models, including deep neural networks. We also propose joint learning to mutually constrain predictions via parameter sharing. In particular, we applied both Bayesian and non Bayesian deep networks using fully connected architectures as well as our own Deep Multi-level Model architecture, which extends the varying-intercepts varying-slopes models which are frequently used for estimation of political opinions in small subgroups. We demonstrate that this method performs well in the classical ecological inference use case, where the analyst is interested in creating cross tabulations of voting patterns by demographic variables of interest, and show that joint training confers significant advantages for modeling the between-race correlations in voting behavior.

Our models are built on the individual level, allowing for individual-level label prediction as well as individual-level latent space modeling. Latent representations trained from ecological models were found to consistently add predictive value over the original feature space on prediction tasks from survey data. The latent space models particularly benefited from the use of deep and Bayesian networks, which resulted in sparser and more predictive representations.

Future work will focus on further developing transfer learning capabilities for building highly predictive models from a combination of ecological and survey data. Since our loss function approximation has increased variance relative to the true loss, tighter approximations based off of sampling methods may increase the signal from data.

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

# A    APPENDIX

| |
|---|
| If the election for U.S. Congress in your district was held today, which one of the following candidates are you most likely to vote for? |
| Generally speaking, do you think of yourself as a Democrat, a Republican, an Independent, or something else? |
| Who would you prefer Joe Biden select as his Vice President for the 2020 Presidential election? |
| If the 2020 presidential election was held tomorrow and the candidates for president were Republican Donald Trump and Democrat Joe Biden , who would you vote for? |
| Which of these statements comes closest to describing your feelings about the Bible? |
| Do you approve or disapprove of the way Donald Trump is handling the economy? |
| If you don't mind saying, who did you vote for in 2016? |
| Do you approve or disapprove of the way Donald Trump is handling the coronavirus pandemic? |
| If you don't mind saying, who did you vote for in 2012? |
| How likely are you to vote in the 2020 presidential election? |
| Are you or someone else in your household a member of a labor union? |
| Generally speaking, would you say that most people can be trusted or that you can't be too careful in dealing with people? |
| Using the following scale, how would you describe yourself politically? |
| How enthusiastic are you to vote in November 2020? |
| Please click on all of the sources of news that you have gotten news from about government and politics in the last week. If you're unsure, please DO NOT click it. – None of these |
| Please click on all of the sources of news that you have gotten news from about government and politics in the last week. If you're unsure, please DO NOT click it. – Twitter |
| Please click on all of the sources of news that you have gotten news from about government and politics in the last week. If you're unsure, please DO NOT click it. – MSNBC |
| Please click on all of the sources of news that you have gotten news from about government and politics in the last week. If you're unsure, please DO NOT click it. – Local television news |
| Please click on all of the sources of news that you have gotten news from about government and politics in the last week. If you're unsure, please DO NOT click it. – Fox News |
| Please click on all of the sources of news that you have gotten news from about government and politics in the last week. If you're unsure, please DO NOT click it. – Broadcast news like ABC, CBS, or NBC |
| Please click on all of the sources of news that you have gotten news from about government and politics in the last week. If you're unsure, please DO NOT click it. – Facebook |
| Please click on all of the sources of news that you have gotten news from about government and politics in the last week. If you're unsure, please DO NOT click it. – CNN |
| Which of these arguments in support and opposition of a Green New Deal do you find most persuasive? |

Table 2: Survey Questions used for Generalization Task

