# OpenReview forum: "Deep Ecological Inference"
_ICLR.cc/2021/Conference — Reject_

### Official Review · AnonReviewer2 · 2020-10-26

**Rating:** 3
**Confidence:** 4

**Review:**

Summary:
This paper uses deep neural networks into the ecological inference problem and shows its effectiveness by using large-scale datasets of the Maryland 2018 midterm elections. Some experimental results have been shown.

Pros.
1. Ecological inference is an important problem in political science for modeling individual-level voting behavior given only aggregate-level data.
2. The authors attempt to apply the recently advanced technique (I.e., DNN) to the ecological inference problem.
3. This paper provides a case study that analyzes real-world voting behavior by using various kinds of datasets.

Cons:
1. The proposed approach is not new.
2. Related work is not adequately cited.
3. This paper is not well-written, especially the Experiment section is not readable.

Reasons for score:
I think the task that the authors address is interesting and important. However, this paper does not present new technical contributions and related works are not listed adequately. The authors attempt to provide the case study of the Maryland 2018 midterm elections via deep learning approaches in ecological settings. But, the Experiment section is not well-organized, and I did not find insightful results from this manuscript. Accordingly, my opinion is that this paper is not ready for publication.

Detailed comments:
1. Important references are missing. So, It is not clear how far the proposed method has pushed the boundary of existing technology. Firstly, as the authors mentioned, the ecological inference is strongly related to the problem “learning with label proportions (LLP)”, the authors should add more discussion.  For example, in [R1], the ecological inference problem has been formulated as LLP. Also, ecological inference relates to other problems discussed in the machine learning community. The common aim is to learn the individual-level models from only aggregate-level data; for example, distribution regression [R2], multiple-instance learning [R3, R4], and collective graphical models [R5, R6]. It would be a great idea to survey and discuss the relationships with these methods.
2. In the approximation procedure for the loss function, the assumption is not clear. Is that “each individual adopts one of the candidates based on the shared probabilities that is the average over $N$ individuals”? If that is the case, the loss is the logarithm of multinomial distribution (i.e., Eq. (4)); this is equivalent to that has been used in the Collective Graphical Models (e.g., [R6]).
3. Section 3.2 is not readable. Does each paragraph in Section 3.2 correspond to which Figure? What is Figure 3.2?
4. In experiments, the authors should specify the experimental settings. For example, what are few-shot settings? Also, the authors should clarify some hyper-parameters such as optimizer selection and learning rate.


     [R1] Tao Sun, Dan Sheldon, Brendan O’Connor, “A probabilistic approach for learning with label proportions applied to the US presidential election”, in ICDM, pp. 445-454, 2017.

     [R2] S. R. Flaxman, Y.-X. Wang, and A. J. Smola, “Who supported Obama in 2012?: Ecological inference through distribution regression,” in Proceedings of the 21st ACM SIGKDD International Conference on Knowledge Discovery and Data Mining. ACM, 2015, pp. 289–298.

     [R3] H. Hajimirsadeghi and G. Mori, “Multi-instance classification by max-margin training of cardinality-based Markov networks,” IEEE Transactions on Pattern Analysis and Machine Intelligence, 2016.

     [R4] H. C. L. Law, D. Sejdinovic, E. Cameron, T. C. D. Lucas, S. Flaxman, K. Battle, and K. Fukumizu. Variational learning on aggregate outputs with Gaussian processes. In NeurIPS, pages 6084–6094, 2018.

     [R5] D. R. Sheldon and T. G. Dietterich, “Collective graphical models,” in Advances in Neural Information Processing Systems, 2011, pp. 1161– 1169.

     [R6] D. Sheldon, T. Sun, A. Kumar, and T. G. Dietterich, “Approximate inference in collective graphical models,” in International Conference on Machine Learning (ICML), vol. 28, no. 3, 2013, pp. 1004–1012.

Minor comments:
- Modify the mathematical symbols to the italic font;  for example, unit $i$ and features $X$ in the sentences of Section 2.2.1.
- Letters in the figures are small and hard to see.

---

### Official Review · AnonReviewer4 · 2020-10-28
**This is an interesting and well-motivated paper that uses techniques from deep learning to generete approximations for the ecological inference problem on voting data. This paper also uses voter file data as an underlying data source, which allows for additional novel insights.**

**Rating:** 7
**Confidence:** 3

**Review:**

This paper proposes a deep learning framework for approximating ecological inference for estimating voting propensities based on demographic aggregates. This is an important problem, as EI has become a court standard for evaluating racially polarized voting in gerrymandering cases for the Gingles factors. Additionally, the increased attention on building coalition districts and availability of individual level data means that this is a problem that is likely to have a large impact in the next redistricting cycle that begins next year.

The proposed methodologies seem natural once the approximation is constructed and this analysis explores some potential ways to incorporate it into various learning architectures. Additional work could be devoted to optimizing over the choices of hyperparameters and providing additional guidance about ways to choose which model would be appropriate based on available input data, since not all applications of these methods will have access to the full sets of surveys and validation measures that were available here. It would be nice to see the performance of these methods on some synthetic data as well and at least one comparison to one of the current state of the art methods on an aggregate version of the data would be useful.

Overall, this paper is interesting and presents an approximation that is likely to be useful in practice for real world problems and given the space constraints appears to present sufficient work to be publishable.


A couple of typos:
Last sentence of paragraph 1 in Section 3 `not correlations' seems like a misnomer

End of caption 1, missing close paren.

---

### Official Review · AnonReviewer3 · 2020-10-28
**Interesting preliminary work, but insufficient experiments/analysis**

**Rating:** 4
**Confidence:** 3

**Review:**

This paper takes an approach to ecological inference inspired by deep learning. Ecological inference is the problem of learning individual labels when only large sets of aggregated data are available. It requires a way to estimate label propensities as a function of covariates. This paper proposes combining a multi-level model with deep learning to estimate voter propensities. The model is then applied to Maryland 2018 midterm election data, and is validated with demographic-level polling data (treating the polling data as ground truth) and with known vote correlations.

Specifically, the paper makes the assumption that distribution of vote counts for a precinct comes from a multinomial distribution, where global probability is averaged across all voters eligible to vote in the precinct (or to a binomial with the same probability averaging). The remaining problem is modeling these probabilities from the individual level covariates. The authors look at 3 models: a linear model, a feed-forward neural network, and a multi-level model where the varying slope coefficients are the output of a feed-forward neural network that takes as input an individual's covariates. The authors consider both frequentist and Bayesian versions of these models. In terms of experimental analysis, the paper argues: the deep models recover the closest demographic splits to the polling estimates, jointly training models across races recover vote covariance structure similar to the truth, and that representations from deep models are useful for few-shot prediction.

I think it's important to connect fields like ecological inference and deep learning. This paper is a solid attempt to combine these disparate fields. The datasets it uses for its experiments are interesting and should be of value to the ML community.

The main problem with the paper is that the experiments are incomplete, unclear, and unconvincing of the significance of the proposed models. For one, the paper doesn't compare to a standard MLM as a baseline -- only the deep MLM, proposed in this paper. How do we know the improvements of a Deep MLM wouldn't be present in the standard method?

Because the main results of this paper are applied, strong analysis is crucial. A high-level issue is that the analysis doesn't go deep enough. There are claims like the following: the deep multi-level model "leads to more reasonable estimates of partisan crossover voting, that is Republican voters voting for Democratic candidates and vice versa, than the dense network, which tends to underestimate the degree of crossover relative to what is found in survey data". Why is this true? Can we interpret the results of the deep MLM that show why this behavior is happening? This is another example where training the regular MLM would be beneficial, so we could compare them and isolate the effects of deep learning. A similar claim is: "Crossover is essentially a matter of capturing the interaction of smaller effects which affect crossover with that of a vary large effect, namely partisanship. We conjecture that the inductive biases of our architecture facilitate the estimation of such interactions." Can we quantitatively show what's happening instead of conjecturing?

Moreover, the model that has the best test set R^2 is the standard linear regression. This is depicted in Table 1 but not discussed in the paper. It is not a make-or-break result, but the paper would benefit from explicitly answering the question: how can we reconcile the fact that the baseline has the best heldout performance but the proposed models perform best on the other tasks?

The paper also omits many key experimental details, which would hamper the reproducibility of results. How are train/test splits created? Are they on the precinct-level? What percent of the data/precinct are in each split? How many hidden units -- and which nonlinearities -- are used for the neural networks? What are the latent features used for the few-shot learning experiment? Is it just the beta's? The beta's concatenated with alphas? Or the entire logits? How do the experiments use the binomial loss for third-party candidates? Is it 1/0 Democrats/not Democrats?

Some smaller comments about the graphs/tables:
  * Where is the mean MSE for the dense baseline in Figure 2C?
  * Why is there no LOO for the 0 hidden layer input for binomial R^2 in Figure 3, i.e. the model input? We only see it for the multinomial.
  * Why is LOO R^2 for baseline dense missing in Table 1?
  * I would remove Figure 1A to clear up space for analysis. Currently, the figure only appears to be showing the fit to the training data, which is not a very salient piece of information.

On a more minor note, there were quite a few typos throughout. Some examples:
  * "represents the candidate the selected" on page 2 [should remove "the"]
  * "This gives us a useful test case for examining not correlations in voting patterns between races" on page 4 [should remove "not"]
  * "We demonstrate two practical use cases for out method" on page 4 ["out" should be "our"]
  * "various depths. performance of the latent space for predicting survey responses." on page 7 [second sentence is incomplete and not capitalized]
  * in general, "R" in "Republican" should be capitalized

In summary, this is an interesting idea and an important research area, but for now it is preliminary work due to incomplete experiments/analysis.

Pros:
- Important research area (combining political science with machine learning)
- Valuable datasets introduced to community
- Model proposed is intuitive

Cons:
- Experiments missing key analysis and baselines
- Experiments aren't reproducible
- Clarity could be improved, typos throughout.

---

### Official Review · AnonReviewer1 · 2020-10-29
**Interesting problem/approach but the paper lacks details and is difficult to follow**

**Rating:** 3
**Confidence:** 3

**Review:**

#### Summary

The paper discusses an interesting direction to efficiently approximate the loss function in the ecological inference problem, which enables extensions using linear models, deep neural networks, and Bayesian neural networks. The proposed approach was evaluated using Maryland 2018 midterm elections data on a range of tasks.

#### Strengths
- The paper tackles one important and practical problem of ecological inference: inferring labels from label proportions, which is applicable to a lot of settings, one of which is "voting" as studied in the paper
- The paper discusses an interesting direction to approximate the loss function of the ecological inference problem in an efficient manner, which enables different extensions, especially using Bayesian neural networks.
- The paper evaluates the proposed model using real Maryland 2018 midterm election data and produces interesting insights

#### Weaknesses
- The paper is not easy to follow. Apart from various typos (see details below), I think the structure of the paper could be improved significantly to make it more accessible. For example:
  - (1) Poisson binomial/multinomial losses were not introduced early in Section 1, which makes it hard to follow and understand the "Contributions" described at the end of Section 1
  - (2) Although described in text in Section 1, it's still pretty unclear what the input data are. I'd suggest discussing the input data formally at the beginning of Section 2 before describing the techniques in details
  - (3) It's very unclear what the evaluation tasks are (especially for people who are not familiar with the data and/or domain) and the intuitions behind why the tasks are suitable to evaluate the effectiveness of the proposed methods

- The paper lacks details on how the proposed methods (and baselines) are implemented. In addition, there are various baselines/methods included in the "results" section but it's unclear what they are in details.

- In addition, there are various typos / minor writing problems. Here are some of them:
  - Sec 3: "This gives us a useful test case for examining not correlations in voting patterns between races." "not correlations"?
  - Sec 3: "We demonstrate two practical use cases for out methods. " -> "We demonstrate two practical use cases for our methods. "
  - References in many places are without parentheses
  - poisson -> Poisson

---

### Decision · Program_Chairs · 2021-01-07
**Final Decision**

**Decision:**

Reject

**Comment:**

This paper is very interesting and timely, but as the reviewers note there is significant room for improvement in the clarity of the presentation and evaluation. In addition to the references mentioned by the reviewers, some other relevant references are the following:

[1] Evan Rosenman, Nitin Viswanathan, "Using Poisson Binomial GLMs to Reveal Voter Preferences," https://arxiv.org/abs/1802.01053

[2] Law, H. C. L., Sutherland, D., Sejdinovic, D., & Flaxman, S. (2018, March). "Bayesian approaches to distribution regression." In International Conference on Artificial Intelligence and Statistics (pp. 1167-1176).